# Relationship Among Body Mass Index, Survival, Cancer Treatment and Health-Related Quality of Life Among Older Patients with Bladder Cancer

**DOI:** 10.3390/cancers17071200

**Published:** 2025-04-01

**Authors:** Mitesh Rajpurohit, Mojgan Golzy, Nai-Wei Chen, Katie S. Murray, Geoffrey Rosen

**Affiliations:** 1Department of Biomedical Informatics, Biostatistics and Medical Epidemiology, University of Missouri at Columbia, Columbia, MO 65201, USA; mrrf9t@health.missouri.edu (M.R.); naiwei.chen@health.missouri.edu (N.-W.C.); 2Department of Urology, NYU-Langone Health, New York, NY 10016, USA; katie.murray@nyulangone.org; 3Department of Urology, Oregon Health & Science University, Portland, OR 97201, USA; geoff.rosen@gmail.com; 4Division of Urology, Department of Surgery, VA Portland Health Care, Portland, OR 97239, USA

**Keywords:** health-related quality of life, activity of daily living, bladder cancer, body mass index, physical and component summary scores

## Abstract

This study explored how body weight (measured by BMI) affects survival, quality of life, and daily activities in older adults with bladder cancer. We analyzed data from over 8000 patients aged 65 and older and found that people with higher BMI (overweight or obese) tended to live longer after bladder cancer diagnosis than those with a normal or underweight BMI. Overweight patients also reported the best overall physical and mental well-being. However, severe obesity came with downsides—these individuals had the most difficulty performing daily tasks like bathing and dressing. Meanwhile, underweight patients also struggled with daily activities. The findings suggest that while extra weight may improve survival and, to some extent, quality of life, extreme obesity can reduce a person’s ability to stay independent. This study highlights the complex relationship between body weight, survival, and well-being in older adults with bladder cancer.

## 1. Background

Bladder cancer (BC) is a disease found mostly in the population aged 55 and above with an average age at diagnosis of 73 [1]. In 2024 alone, it was estimated to have affected 83,190 individuals in the United States and led to 16,840 deaths [2]. Following a BC diagnosis, patients often experience a significant decline in functional status and overall health compared to their pre-diagnosis state [3]. Poor general health and impaired ability to perform activities of daily living (ADLs) are among the strongest risk factors for depression in older BC patients [4]. The association between BC outcomes and body mass index (BMI) is widely researched. A 2023 umbrella review showed that being overweight and underweight were positively and negatively associated with increased incidences of BC, respectively [5,6].

Various studies have reported the effect of BC on activities of daily living (ADLs). A 2017 systematic review showed that almost half of those who are treated with radical cystectomy are frail or prefrail [7]. These individuals more frequently develop postoperative adverse events such as major complications and early mortality. Another study by Monfardini et al. found that more than 55% of patients with bladder and renal cancer had at least one Cumulative Illness Rating Scale grade three (severe) or four (extremely severe) comorbidity [8]. Persons affected by BC often have smoking exposure, high BMI, and are insufficiently active [9]. Our previous study demonstrated that patient clusters based on mental and physical functioning exhibit significantly different survival outcomes, independent of BC disease severity or treatment type [10]. Additionally, within two years of diagnosis, a quarter of BC patients experience substantial changes in quality of life [11]. The association between body mass index (BMI) and BC survival is not clear, with some studies showing better outcomes and others showing worse outcomes [12]. Some studies have found that patients with higher BMI had better survival within the first five years after radical cystectomy. Other studies have found that patients with higher BMI had better outcomes [13,14,15].

There is also considerable interest in the relationship between BMI and Health-Related Quality of Life (HRQoL) in BC survivors across settings [16,17,18,19]. Generally, a higher BMI is consistently linked to lower HRQoL, highlighting the negative impact of obesity on survivors’ well-being [20]. However, the studies also reveal variability in the magnitude of these effects, pointing to the influence of demographic, clinical, and treatment-related variables.

This study aims to clarify the relationship between BMI, HRQoL, and survival in BC patients aged 65 and older. Herein, we will (1) evaluate the impact of BMI on health-related quality of life (HRQoL) in patients with BC and (2) assess the association between BMI and survival in these patients.

## 2. Methods

Study Design: This is a population-based retrospective cross-sectional cohort study conducted using the Surveillance, Epidemiology, and End Results-Medicare Health Outcomes Survey (SEER-MHOS).

Study Population: This study utilized data from the SEER-MHOS database spanning 1999 to 2021 [21,22]. SEER-MHOS offers a unique opportunity for cancer research by integrating cancer-specific data with health-related quality of life (HRQOL) information for Medicare Advantage enrollees. This large, longitudinal dataset enables the investigation of long-term cancer impacts on HRQOL, encompassing physical, emotional, and social well-being, as well as the reverse relationship.

Our study focused on older patients with bladder cancer (BC). We included individuals aged ≥ 65 years who had a documented BC diagnosis and a recorded body mass index (BMI), our primary variable of interest. To ensure a more homogeneous study population, we excluded patients with a survival duration of less than six months, minimizing potential confounding from individuals experiencing rapid mortality.

For patients with multiple records, we selected the most recent survey as the baseline to ensure consistency in follow-up calculations for survival analysis. Given that some patients had recurrent BC diagnoses, older survey records—some dating back as far as a decade—might not accurately represent their current health status. Therefore, we prioritized the most recent survey to capture the latest clinical information, enhancing the relevance and reliability of our analysis.

Data Quality: The SEER-MHOS dataset maintains high data quality through rigorous auditing and validation processes. These measures ensure that the resource remains a reliable and valuable tool for cancer research. SEER-MHOS spans over two decades (1999–2021), providing a large sample for analyzing long-term trends. Additionally, SEER captures approximately 28% of the U.S. population, offering comprehensive cancer surveillance data with demographic diversity [23]. SEER-MHOS maintains high data quality through rigorous data collection protocols, periodic audits, and standardized validation processes. SEER follows strict case ascertainment procedures, while MHOS ensures self-reported HRQOL measures are collected consistently across survey cycles [22]. The linkage between SEER and MHOS allows researchers to study cancer patients’ health outcomes with high reliability.

Outcome Measures: Outcome variables of interest included (1) HRQoL measured by the physical component summary (PCS), and mental component summary (MCS), (2) overall ability to perform ADLs (ADLs), and (3) survival.

The physical component summary (PCS) is a comprehensive measure of physical health, derived from responses across multiple domains, including physical functioning, role limitations due to physical health, bodily pain, and general health perceptions. It is a summary score ranging from 0 to 100, with a higher score indicating better physical health. The mental component summary (MCS) is a comprehensive measure of mental health, derived from responses across multiple domains, including general mental health, role limitations due to emotional problems, social functioning, and vitality. Like the PCS, it is a summary score ranging from 0 to 100, with a higher score indicating better mental health [24]. ADLs consist of bathing, dressing, eating, sitting on/standing up from a chair, walking, and using the toilet. The overall ADL ability was classified into three categories: “No difficulty in any ADL”, “Difficulty in at least one ADL”, and “Disability in at least one ADL”.

Independent Variable: The primary independent variable was BMI (kg/m^2^), with the following classifications: BMI < 18.5 as “Underweight”; 18.5 ≤ BMI < 25 as “Normal weight”; 25 ≤ BMI < 30 as “Overweight”; 30 ≤ BMI < 40 as “Obesity”; and 40 ≤ BMI as “Severe obesity”.

Covariates: Additional baseline characteristics (including age, gender, race, marital status, home ownership, education level, income, and smoking status); cancer characteristics (including surgery, chemotherapy, and stage of cancer); comorbidities (including muscular disease, depressive symptoms for more than two weeks of the last year, pre-existing comorbid health conditions [computed as a summed disease burden of self-reported physician diagnosed hypertension, cardio-vascular disease, chronic pulmonary obstruction disease, diabetes mellitus, gastrointestinal disease, stroke, and other cancer]); and time from the first cancer diagnosis to the survey.

Statistical Analysis: Patient characteristics at the baseline were summarized with descriptive statistics and presented as frequencies (percentages) for categorical variables and means (standard deviations) for continuous variables in the text and table. Bivariate analyses were stratified by the level of BMI categories. The chi-square test of independence was used to assess the association of the categorical outcome (e.g., ADL) and BMI categories. Analysis of variance (ANOVA) was used to assess the association of numeric outcomes (e.g., HR-QoL, PCS, and MCS) and BMI categories. All assumptions were inspected, and a non-parametric test such as Kruskal–Wallis was used when the normality assumption was violated [25].

To further evaluate the effect of BMI categories on HR-QoL (e.g., PCS and MCS), a generalized linear model (GLM) was used, adjusting for covariates [26]. Covariates in the GLM model included age, gender, race, income, home ownership, education, smoking status, surgery type, chemotherapy, cancer stage, pre-existing depression symptoms, muscular disease, pre-existing comorbid health conditions, and time from the first cancer diagnosis to the survey. Residual analysis was applied to check assumptions in linear regression. To account for the skewness in the MCS, the Box–Cox transformation (MCS_cox_ = [MCS^λ^−1]/λ) was used with optimal lambda (λ) = 2.25 [27]. The effect of BMI categories was evaluated on the transformed the MCS in a multivariable GLM. Additionally, a log-rank test corresponding to Kaplan–Meier survival analysis was used to assess whether there was any significant difference in survival probability across BMI categories [28,29]. All tests of statistical significance were two-sided with a significance level of 0.05. Analyses were performed with SAS version 9.4 (SAS Institute, Inc., Cary, NC, USA) [30].

## 3. Results

We included 33,145 SEER-MHOS records of 18,827 patients with BC (Figure 1). We excluded 14,259 records prior to BC diagnosis, 565 records prior to age 65, and 539 records with less than 6 months of post-survey survival. We excluded 5001 records without BMI data. This left us with 12,781 surveys related to 8013 patients. We only included the last survey from each patient, leaving us with 8013 surveys from an equal number of patients.

The mean age at the time of survey of the sample set was 77.68 years with a standard deviation of 7.08 years. Most of the participants were male (74.8%) and White (85.6%). The majority were married (58.7%), homeowners (74.5%), high school graduates or less educated (54.7%), earning an annual income of USD 20,000 to 49,000 (37.9%), non-smokers (85.7%), and in the overweight BMI category (40.7%) (Table 1).

### 3.1. Cancer Charateristics and BMI Association

The time from diagnosis to survey completion was correlated with time since cancer diagnosis, with the shortest interval for those with severe obesity and the longest for underweight individuals (*p* < 0.001, Table 2). Survival was also correlated with BMI (*p* < 0.001), with shortest survival for severe obese and underweight patients and longest survival for overweight patients.

Months since first cancer diagnosis to survey reports the time elapsed from the initial diagnosis to study inclusion, regardless of potential disease-free intervals.

The cancer stage at diagnosis was not correlated with BMI at survey completion (*p* = 0.17, Table 3). Similarly, there was no statistically significant correlation between BMI and the type of surgical treatment (*p* = 0.24) or receipt of chemotherapy (*p* = 0.57).

### 3.2. Physical and Mental Component Summary Scores and BMI Association

The overall mean PCS score in the sample population was 37.3 ± 12.1, with the highest score in the overweight category (38.5) and the lowest in the severe obesity category (30.7). The overall mean MCS score was 51.6 ± 11.4, with the highest mean observed in the overweight category (52.3) and the lowest in the severe obesity category (46.5) (Figure 2).

The results of ANOVA showed a significant difference in PCS scores among the BMI categories (F_(4,7495)_ = 32.66, *p* < 0.0001). The distribution of the MCS scores was right-skewed, and so the nonparametric Kruskal–Wallis (KW) test was performed for univariate analysis of the MCS scores, indicating a significant difference in distribution of the MCS among the BMI categories (KW *p*-value < 0.0001).

When looking at the PCS and MCS scores over time from 2007 to 2019, we observed a trend towards higher scores (*p* < 0.0001, Figure 3) for the MCS, signifying improved mental health over time. No significant overall change in the PCS scores was observed (*p* = 0.9). From 2007 to 2019, we observed a stable PCS score for the healthy-range, overweight, and obesity subgroups over time, a trend towards higher scores for the underweight subgroup, and a trend towards lower scores for sever obese subjects over time.

ADLs and BMI Association: There was a significant difference in ADL ability correlated with BMI (*p* < 0.0001, Table 4). Severely obese individuals had the highest percentage of individuals with at least one ADL difficulty (57.8%) and the highest for disabilities (18.5%), while underweight and obese patients had similar levels of overall disability (45.4% having difficulty with one ADL and no disability for both). Healthy-range and overweight individuals had fewer ADL difficulties and disabilities, with a majority having no difficulties.

### 3.3. Survival Analysis by BMI

Kaplan–Meier survival analysis demonstrated that the median survival time after survey completion was 81.8 months. Our findings highlight a significant variability in survival outcomes among different BMI categories, with underweight and normal-weight individuals experiencing the worst survival (log-rank test *p* < 0.0001, Figure 4).

We compared the KM survival curves among BMI groups with first BC diagnosis as the baseline, and we observed a similar result which shows the consistency of the result. Figure 5, rather than capturing uninterrupted disease duration, reports the time elapsed from the initial diagnosis to study inclusion, regardless of potential disease-free intervals.

### 3.4. Multivariable Regression Analysis of PCS and MCS Outcomes

In multivariate analysis, with the normal range BMI as the reference, underweight, and severe obesity had significantly lower PCS and MCS scores (Table 5). Obese patients had significantly lower PCS scores, and overweight patients had significantly better mental health than the normal range BMI. Modeling results also revealed that age is negatively associated with both the PCS and MCS results, indicating that older age results in lower scores. Race and gender were not associated with the PCS, but females and White participants had a significantly better MCS outcome. Variables such as depression symptoms, non-house ownership, low income, and the number of comorbidities were linked to significantly lower PCS and MCS scores. Smoking, muscular diseases, and undergoing cystectomy are also linked to significantly lower PCS scores. The bladder cancer stage, surgical intervention, receipt of intravenous chemotherapy, and time from the first cancer diagnosis to the survey were not significantly associated with the MCS outcome. The time from the first cancer diagnosis to the survey has no significant effect on either the PCS (*p* = 0.4066) or MCS scores (*p* = 0.9082).

## 4. Discussion

Our final cohort data are in line with prior studies both in terms of age [1]. sex [31], and race [32]. Our study highlights the complex interplay between BMI, demographic factors, and patient-reported HRQoL outcomes in older patients with BC. The significant association between obesity and lower physical component summary (PCS) scores aligns with the previous literature, suggesting that excess body weight may contribute to physical limitations and reduced mobility, ultimately impacting overall physical well-being [20]. Our study indicates low PCS scores for underweight individuals as well. Interestingly, overweight individuals demonstrated significantly better mental component summary (MCS) scores compared to those with normal-range BMI, which may indicate potential psychological resilience or benefits associated with a slightly higher body weight [33].

Age was a determinant of both physical and mental health, with increasing age negatively associated with PCS and MCS scores. This finding is expected, as aging is often accompanied by a decline in physical function, increased comorbidities, and potential psychosocial challenges, such as reduced social engagement and loss of independence.

While race and gender did not significantly impact PCS scores, females and White individuals had significantly higher MCS scores. This may reflect differences in social support, healthcare access, coping mechanisms, or cultural perceptions of mental health, warranting further exploration into underlying psychosocial or systemic factors.

Socioeconomic and health-related variables played a significant role in both PCS and MCS outcomes. Depression symptoms, lower income, lack of home ownership, and a higher number of comorbidities were strongly linked to poorer HRQoL across both dimensions. These results underscore the importance of addressing social determinants of health in improving patient well-being. Additionally, smoking, muscular diseases, and undergoing cystectomy were associated with lower PCS scores, suggesting that these conditions contribute to significant physical health burdens.

The time from the first cancer diagnosis to the survey has no significant effect on either the PCS or MCS scores.

Notably, cancer-related variables such as cancer stage, surgery type, chemotherapy, and time from the first cancer diagnosis to the survey did not significantly impact MCS outcomes. This finding suggests that while cancer treatment can impose physical stress, mental health outcomes may be more influenced by pre-existing conditions, coping mechanisms, and social support rather than the specific cancer treatment characteristics.

Our findings also highlight a significant variability in survival outcomes among different BMI categories, with underweight and normal-weight individuals experiencing the worst survival. This aligns with prior research suggesting that low BMI may be associated with higher frailty, reduced physiological reserves, and increased vulnerability to disease-related complications [34]. The poor survival in the underweight group may reflect underlying malnutrition, muscle wasting (sarcopenia), or more advanced disease states at original diagnosis.

Interestingly, overweight and obese individuals demonstrated better survival outcomes [15]. Several other studies also reported that higher BMI is associated with better survival in bladder cancer patients [35,36]. While obesity is typically associated with higher risks of chronic diseases, higher BMI in bladder cancer patients may provide protective benefits. These may include greater metabolic reserves during periods of severe illness, better tolerance to treatment-related stress, and differences in inflammatory responses. However, it remains unclear whether these benefits are directly due to excess body weight or other underlying factors such as muscle mass, nutritional status, or differences in treatment received.

We compared Kaplan–Meier (KM) survival curves among BMI groups using two different baseline time points: the time of the survey (Figure 4) and the first cancer diagnosis (Figure 5). This analysis aimed to identify any inconsistencies between the approaches. Our findings demonstrated a consistent pattern across both methods. In this dataset, the average time from the first bladder cancer (BC) diagnosis to the most recent survey (i.e., inclusion in the study) was 8.5 years (102 months). However, it is important to acknowledge that BC can recur following initial treatment. For patients with recurrent BC, there may have been periods of disease-free survival between episodes of recurrence.

### 4.1. Limitations

This study has several limitations that should be considered when interpreting the findings. In terms of sample selection, our analysis was based on data obtained from the SEER–MHOS database, which comprises survey information on bladder cancer patients enrolled in the Medicare Advantage Program from participating registries. As a result, the study population may not fully represent the diverse demographics of the broader U.S. population, nor can the findings be generalized to other countries. Nonetheless, our data are in line with prior studies both in terms of age [1], sex [31], and race [32].

As BMI was the main variable of interest, we excluded individuals with missing BMI, which may introduce the potential for selection bias if the missingness is not random, as these individuals may differ systematically from those included in the analysis. This study occurred over a long timeframe over which the management of bladder cancer significantly changed, and it continues to do so.

Regarding study design, the cross-sectional nature of this study limits its ability to establish causality and exploration of temporal relationships between BMI, physical and mental component summary (PCS and MCS) scores, and activities of daily living (ADLs) disabilities. Consequently, causal inferences or longitudinal insights into these associations cannot be drawn from the current findings.

### 4.2. Strengths of the Study

The SEER-MHOS data resource is one of the most comprehensive resources available for analyzing cancer patients. It provides a unique opportunity for cancer research by combining cancer-specific data with health-related quality of life (HRQOL) information for Medicare Advantage enrollees. This is a unique and large dataset that gives researchers the opportunity to study the long-term impact of cancer on health-related quality of life, including physical, emotional, and social well-being. Additionally, it can be used to compare cancer survival among different populations.

## 5. Conclusions

Our study underscores the multifaceted relationship between BMI, demographic and socioeconomic factors, and HRQoL outcomes, as well as survival disparities among BMI categories for older patients with bladder cancer. Future research should aim to further explore the mechanisms driving these associations and investigate targeted interventions to improve HRQoL and survival outcomes, particularly for high-risk groups such as underweight individuals and those with socioeconomic disadvantages. Integrating personalized treatment strategies that consider both physical and mental health factors could lead to better overall patient outcomes and quality of life.

## Figures and Tables

**Figure 1 cancers-17-01200-f001:**
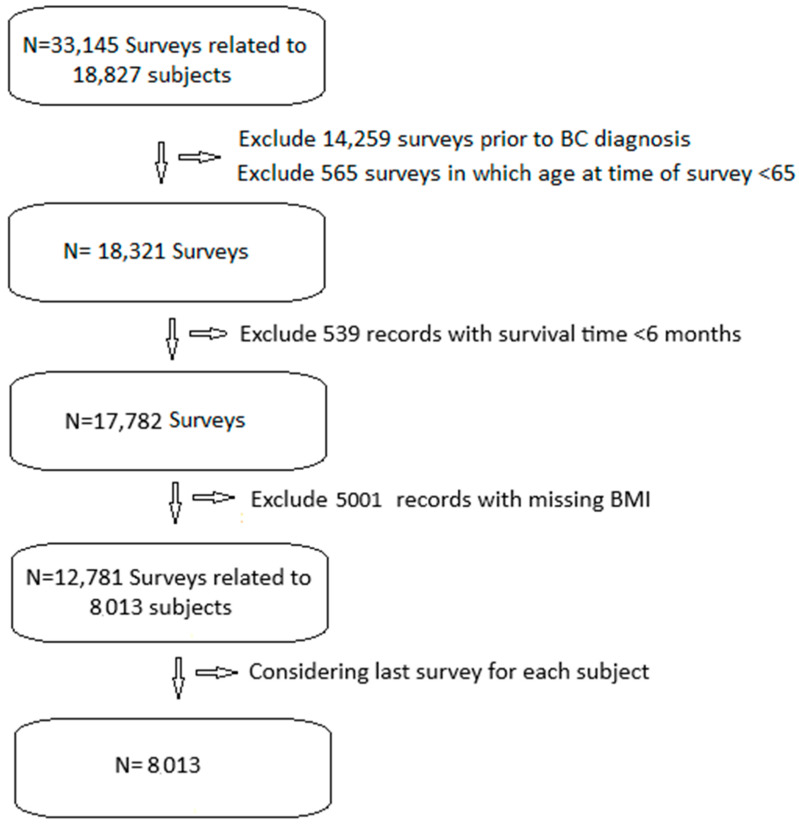
Study flow diagram.

**Figure 2 cancers-17-01200-f002:**
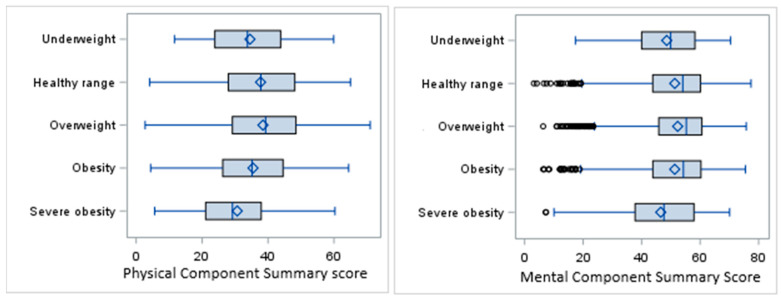
Box plots for PCS and MCS scores by BMI categories.

**Figure 3 cancers-17-01200-f003:**
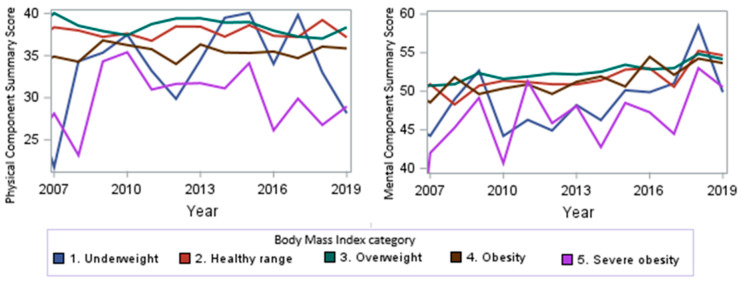
PCS and MCS score trends by year.

**Figure 4 cancers-17-01200-f004:**
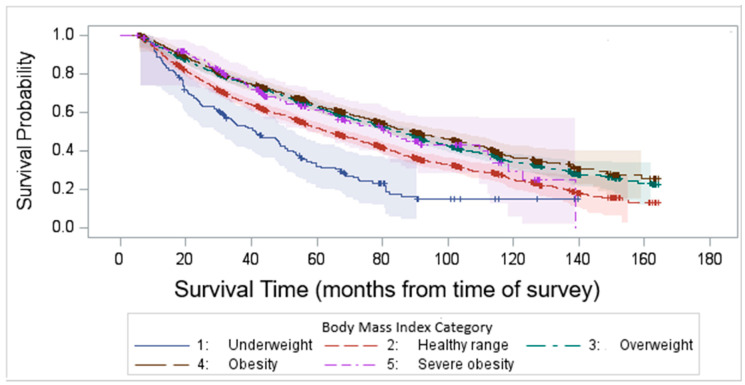
Kaplan–Meier survival time after survey completion by BMI categories.

**Figure 5 cancers-17-01200-f005:**
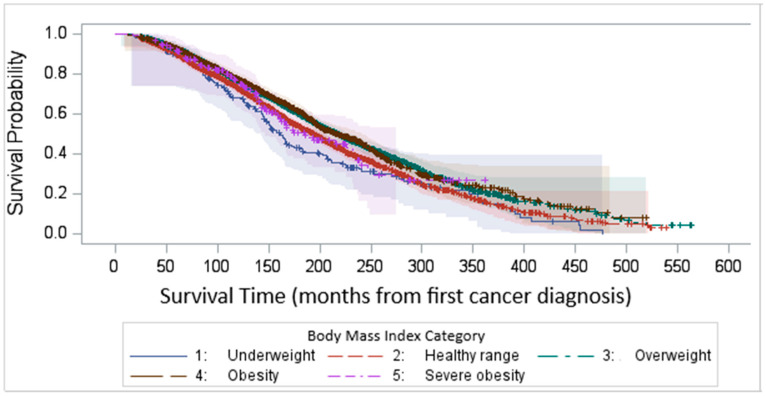
Kaplan–Meier survival time after first cancer diagnosis by BMI categories.

**Table 1 cancers-17-01200-t001:** Summary statistics of demographic and socioeconomic variables.

Variables	Level	Overall (N = 8013)
Age at survey Mean (SD)		77.68 (7.08)
Gender: *n* (%)	Female	2017 (25.2%)
	Male	5996 (74.8%)
Marital status: *n* (%)	Married	4706 (58.7%)
	Other	3237 (40.4%)
	Unknown	70 (0.9%)
Home Ownership: *n* (%)	Owner	5966 (74.5%)
	Other	1803 (22.5%)
	Unknown	244 (3.0%)
Race: *n* (%)	White	6860 (85.6%)
	Hispanic	463 (5.8%)
	Black or African American	345 (4.3%)
	Asian or Pacific Islander	309 (3.9%)
	American Indian or Alaskan native	25 (0.3%)
	Another race or multi-race	11 (0.1%)
Education: *n* (%)	2–4 years	2640 (33.0%)
	Above 4 years	861 (10.7%)
	High school or less	4380 (54.7%)
	Unknown	132 (1.6%)
Income: *n* (%)	Less than USD 20,000	2026 (25.3%)
	USD 20,000—49,999	3036 (37.9%)
	USD 50,000 or more	1490 (18.6%)
	Unknown	1461 (18.2%)
Smoking status: *n* (%)	Yes	1025 (12.8%)
	Not at all	6865 (85.7%)
	Don’t know or missing	123 (1.5%)
Body mass index group, kg/m^2^: *n* (%)	Underweight (BMI < 18.5)	181 (2.3%)
	Healthy range (18.5 ≤ BMI < 25)	2638 (32.9%)
	Overweight (25 ≤ BMI < 30)	3258 (40.7%)
	Obesity (30 ≤ BMI < 40)	1756 (21.9%)
	Severe Obesity (40 ≤ BMI)	180 (2.2%)

**Table 2 cancers-17-01200-t002:** Mean and standard deviation of time in months since first cancer diagnosis to survey and survival time in months since time of survey.

	Overall	Underweight	Healthy	Overweight	Obesity	Severe Obesity
Months since first cancer diagnosis to survey (*p* < 0.001)	102 ± 84	118 ± 94	107 ± 88	101 ± 83	96 ± 79	87± 70
Survival time since survey (*p* < 0.001)	57 ± 37	41 ± 29	53 ± 37	60 ± 38	59 ± 38	53 ± 33

**Table 3 cancers-17-01200-t003:** Distribution of cancer characteristics overall and by BMI categories.

Variables	Overall	Underweight	Healthy	Overweight	Obesity	Severe Obesity
Stage (*p* = 0.17)						
Non-muscle invasive	6519 (81%)	149 (82%)	2128 (81%)	2632 (81%)	1461 (83%)	149 (83%)
Muscle invasive	1196 (15%)	21 (12%)	405 (15%)	509 (16%)	234 (13%)	*
Metastatic	298 (3.7%)	*	105 (4.0%)	117 (3.6%)	61 (3.5%)	*
Surgery (*p* = 0.24)						
Cystectomy	980 (12%)	20 (11%)	343 (13%)	413 (13%)	187 (11%)	17 (9%)
Transurethral	3722 (46%)	86 (48%)	1230 (47%)	1526 (47%)	802 (46%)	88 (49%)
None/unknown	3311 (41%)	75 (41%)	1065 (40%)	1329 (41%)	767 (44%)	75 (42%)
Chemotherapy (*p* = 0.57)						
Yes	969 (12%)	17 (9%)	310 (12%)	414 (13%)	208 (12%)	20 (11%)
None/unknown	7044 (88%)	164 (91%)	2328 (88%)	2844 (87%)	1548 (88%)	160 (89%)

* Numbers are masked per CMS policy.

**Table 4 cancers-17-01200-t004:** Frequency and column percentages of overall ADL ability by BMI categories.

Ability to Perform ADLs * (*p* < 0.0001)	Underweight	Healthy Range	Overweight	Obesity	Severe Obesity
No difficulty in any ADLs	77 (44.3%)	1525 (59.3%)	1926 (60.8%)	838 (48.5%)	41 (23.7%)
Difficulty in at least one ADL	79 (45.4%)	867 (33.7%)	1103 (34.8%)	784 (45.4%)	100 (57.8%)
Disability in at least one ADL	18 (10.3%)	180 (7%)	141 (4.4%)	106 (6.1%)	32 (18.5%)

* Frequency missing = 196.

**Table 5 cancers-17-01200-t005:** Estimates of effect of patient characteristics on PCS and transformed MCS outcomes in multivariable GLMs.

	PCS Outcome			Transformed MCS Outcome		
Variables	Estimate	SE	*p* Value	Estimate	SE	*p* Value
BMI Group (reference: Healthy range)						
Underweight	−1.74	0.89	0.05	−199.88	104.24	0.0552
Overweight	0.34	0.30	0.2541	76.69	35.50	0.0308
Obesity	−2.15	0.36	<0.0001	27.40	42.66	0.5208
Severe obesity	−5.72	0.91	<0.0001	−321.00	106.35	0.0026
Age	−0.30	0.02	<0.0001	−5.68	2.29	0.0131
Time from first cancer diagnosis	0.00	0.00	0.4066	0.02	0.18	0.9082
Gender (reference: Male)						
Female	−0.36	0.30	0.2335	78.99	35.25	0.0251
Race (reference: White)						
Black or American African	0.56	0.64	0.3853	−256.98	75.43	0.0007
Other	−0.03	0.44	0.9536	−335.63	51.21	<0.0001
Depression (reference: No)						
Yes	−2.46	0.37	<0.0001	−1351.7	43.28	<0.0001
Muscular disease (reference: No)						
Yes	−0.51	0.33	0.1245	79.14	39.28	0.044
Surgery (reference: None/Unknown)						
Cystectomy	−1.07	0.48	0.0257	−25.02	56.47	0.6578
Transurethral	−0.31	0.29	0.2723	−30.89	33.47	0.356
Chemotherapy (reference: No)						
Yes	−0.43	0.41	0.2985	27.04	48.59	0.5779
Stage (reference: Non-muscle invasive)						
Metastatic	−0.18	0.70	0.7995	−55.37	81.70	0.498
Muscle invasive	0.19	0.41	0.6396	−39.42	48.29	0.4144
Income (reference: USD 20,000-49,999)						
≥USD 50,000	1.11	0.37	0.0028	137.17	43.73	0.0017
<USD 20,000/Unknown	−0.84	0.29	0.0044	−126.47	34.56	0.0003
Home Ownership (reference: Owner)						
Other	-0.98	0.31	0.0015	−205.11	36.09	<0.0001
Education (reference: High school or less)						
2–4 years	0.72	0.29	0.0128	104.81	33.76	0.0019
>4 years	2.07	0.45	<0.0001	125.71	52.75	0.0172
Smoker (reference: None/unknown)						
Yes	−2.08	0.39	<0.0001	−43.75	45.97	0.3413
Number of comorbidities (reference: 2 or fewer)						
≥5	−5.23	0.34	<0.0001	−153.73	39.57	0.0001
3 or 4	−11.07	0.42	<0.0001	−513.70	49.60	<0.0001

## Data Availability

Data for this study were obtained from the SEER-MHOS link data resource. The SEER-MHOS database is available to outside investigators for research purposes (please see https://healthcaredelivery.cancer.gov/seer-mhos/obtain/overview.html) (accessed on 31 March 2025).

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
