# Peer review of "Relationship Among Body Mass Index, Survival, Cancer Treatment and Health-Related Quality of Life Among Older Patients with Bladder Cancer"

_cancers, 2025, doi:10.3390/cancers17071200_

Round 1
Reviewer 1 Report
Comments and Suggestions for Authors
Thank you for the opportunity to review manuscript ID: cancers-3524701. This manuscript aimed to analyse whether Body Mass Index (BMI) was associated with health-related quality of life (HRQoL) and survival in patients over 65 with a diagnosis of bladder cancer (BC).
In my opinion the paper is not satisfactorily clear or informative.
The Introduction section satisfactorily presented the rationale of this study. The knowledge gap on this topic is well defined. The aim of this study is well expressed and of interest.
Major comments:
Methods section
- Subsection `Study design` is missing. Correct this (with a detailed description).
- Subsection `Data quality` is missing (for example, coverage, completeness, accuracy, reliability). Correct this (with a detailed description). Cite appropriate references.
- Define and describe `Outcome variables of interest included 1) HRQoL measured by the physical component summary (PCS), and mental component summary (MCS), 2) overall ability to perform ADL (ADLs)`, and cite appropriate references.
- Outcome Measures: Match the variables in the Methods section and in Table 4, that is, I quote
“No difficulty in any ADL”,
“Difficulty in at least one ADL”, and
“Disability in at least one ADL”.
with
`No difficulty in any ADL`
`At least one difficulty but no disability`
`At least one ADL disability`.
In the Discussion section, the results of this manuscript are unsatisfactorily compared and discussed in context with the results of some other similar studies, with citing only one reference in the entire text of the Discussion. Correct this.
In particular, explain the effect of the duration of the disease on the results of this study.
The subsection Limitations should be supplemented with other shortcomings of this work (selection bias, quality data, some other variables that are not included in this study, etc.). Discuss the possibilities for overcoming the shortcomings of this manuscript.
The conclusions are generally supported by the results.
Author Response
Thank you for reviewing our manuscript and your valuable and constructive comments.
Comment 1- Subsection `Study design` is missing. Correct this (with a detailed description).
Response: Thank you for your valuable comment. We have added a Study Design subsection to provide a detailed description of our methodological approach.
This study is a population-based, retrospective, cross-sectional cohort study utilizing data from the SEER-MHOS resource. Specifically, it:
Population-based: Focuses on individuals diagnosed with bladder cancer (BC) within the SEER-MHOS dataset.
Retrospective: Analyzes pre-existing data, looking back in time to examine patient characteristics and outcomes.
Cross-sectional: Uses the most recent survey for each patient as the baseline, capturing HRQOL and clinical factors at a single time point.
Cohort approach: Follows individuals over time to assess survival outcomes while considering prior diagnoses and disease progression.
This design allows us to investigate long-term impacts of BC on health-related quality of life and survival while leveraging the strengths of a large, longitudinal dataset.
Comment 2- Subsection `Data quality` is missing (for example, coverage, completeness, accuracy, reliability). Correct this (with a detailed description). Cite appropriate references.
Response: Thank you for your question. We have added a Data Quality subsection to provide a comprehensive description of the dataset’s coverage, completeness, accuracy, and reliability.
The SEER-MHOS data resource, used in this study, integrates cancer registry data from the Surveillance, Epidemiology, and End Results (SEER) program with health-related quality of life (HRQOL) information from the Medicare Health Outcomes Survey (MHOS). This dataset is designed to enhance our understanding of the long-term impacts of cancer diagnosis and treatment on patient well-being.
Coverage and Completeness:
SEER-MHOS includes a nationally representative sample of Medicare Advantage beneficiaries diagnosed with cancer, ensuring broad population coverage. The dataset spans over two decades (1999–2021), providing a large sample for analyzing long-term trends. Additionally, SEER captures approximately 28% of the U.S. population, offering comprehensive cancer surveillance data with demographic diversity (NCI, 2023).
Accuracy and Reliability:
SEER-MHOS maintains high data quality through rigorous data collection protocols, periodic audits, and standardized validation processes. SEER follows strict case ascertainment procedures, while MHOS ensures self-reported HRQOL measures are collected consistently across survey cycles (Ambs et al., 2008; NCI, 2023). The linkage between SEER and MHOS allows researchers to study cancer patients' health outcomes with high reliability.
Justification for Study Design and Patient Selection:
Studying rare diseases, such as cancer, presents challenges in assembling sufficiently large cohorts through prospective studies. Retrospective analyses using existing datasets like SEER-MHOS offer several advantages, including:
Larger sample sizes, improving statistical power.
Cost-effective analyses, reducing the burden of primary data collection.
Faster study completion, facilitating timely research on cancer outcomes.
To enhance transparency, we have detailed patient exclusion criteria in the manuscript and clarified our selection process. Despite some inherent limitations, our final cohort is consistent with prior studies in terms of key demographic factors such as age, sex, and race, reinforcing the dataset’s representativeness.
References:
Ambs A, Warren JL, Bellizzi KM, et al. Overview of the SEER-MHOS linked data resource. Med Care. 2008;46(8 Suppl 1):S1-S6.
National Cancer Institute (NCI). SEER-MHOS: Data Documentation and Methods. 2023. Available at: https://healthcaredelivery.cancer.gov/seer-mhos/aboutdata/documentation.html
Comment 3- Define and describe `Outcome variables of interest included 1) HRQoL measured by the physical component summary (PCS), and mental component summary (MCS), 2) overall ability to perform ADL (ADLs)`, and cite appropriate references.
Response: Thank you for your question. Outcome measures subsection is revised to include describe outcome variables.
- Health related quality of life (HRQoL) measured by the physical component summary (PCS), and mental component summary (MCS).
The Physical Component Summary (PCS) is a comprehensive measure of physical health, derived from responses across multiple domains including physical functioning, role limitations due to physical health, bodily pain, and general health perceptions. It is a summary score ranging from 0 to 100, with a higher score indicating better physical health.The Mental Component Summary (MCS) is a comprehensive measure of mental health, derived from responses across multiple domains including general mental health, role limitations due to emotional problems, social functioning, and vitality. Like PCS, it is a summary score ranging from 0 to 100, with a higher score indicating better mental health (23).
2) overall ability to perform ADL (ADLs)
ADLs consists of bathing, dressing, eating, getting in/out of the chair, walking, and using the toilet. The overall ADL ability was classified into three categories: “No difficulty in any ADL”, “Difficulty in at least one ADL”, and “Disability in at least one ADL”.
The following reference is added
Ware, JE, Sherbourne CD. “The MOS 36-Item Short-Form Health Survey (SF-36): I. Conceptual Framework and Item Selection.” Medical Care, vol. 30, no. 6, 1992, pp. 473–83. JSTOR, http://www.jstor.org/stable/3765916. Accessed 25 Mar. 2025.
Comment 4- Outcome Measures: Match the variables in the Methods section and in Table 4, that is, I quote
“No difficulty in any ADL”,
“Difficulty in at least one ADL”, and
“Disability in at least one ADL”.
with
`No difficulty in any ADL`
`At least one difficulty but no disability`
`At least one ADL disability`.
Response: Thank you for your comment. It is fixed.
Comment 5- In the Discussion section, the results of this manuscript are unsatisfactorily compared and discussed in context with the results of some other similar studies, with citing only one reference in the entire text of the Discussion. Correct this.
In particular, explain the effect of the duration of the disease on the results of this study.
Response: Thank you for your question. Citations are added in the discussion section. The discussion regarding the duration of disease is also added the discussion section.
In this data set, on average, the time from the first bladder cancer (BC) diagnosis to the most recent survey (i.e., inclusion in the study) was 8.5 years (102 months). However, it is important to note that BC can recur after initial treatment. For patients with recurrent BC, there may be periods during which they were disease-free between episodes of recurrence. As a result, we cannot precisely determine the continuous duration a patient lived with BC. Instead, our reported value reflects the time from initial diagnosis to study inclusion, irrespective of potential disease-free intervals.
However, we have compared the long-term survivorship among BMI groups, and we observed similar results with underweight group having the worst survival outcome.
Regarding the effect of duration of disease on PCS and MCS, we did not find any significant effect of time from first cancer to survey on PCS (p=0.4066) and MCS (p=0.9082).
Comment 6-The subsection Limitations should be supplemented with other shortcomings of this work (selection bias, quality data, some other variables that are not included in this study, etc.). Discuss the possibilities for overcoming the shortcomings of this manuscript.
Response: Thank you for your question. We have revised the Discussion section by incorporating additional citations to better contextualize our findings in relation to prior studies. Additionally, we have expanded the discussion regarding the effect of disease duration on our results.
Thanks again for our your valuable comments.
Reviewer 2 Report
Comments and Suggestions for Authors
Dear Authors,
Congratulations on your work, I believe the topic is of high importance. Nevertheless, I have a few suggestions for improving the manuscript and making it suitable for publishing.
- Please explain why so many patients have been excluded from the study starting so early.
- Why has only the last survey for each patient been investigated? What was the average time the patients lived with bladder cancer before their inclusion in the study? As cancer represents a dynamic and ofter unpredictable disease, I believe the Authors should improve the materials and methods, as well as the results sections of the manuscript, particularly by offering greater transparency when it comes to the selection of the patients, as well as their baseline characteristics.
- Please state the aim and objectives of your study more clearly at the end of the Introduction section.
Thank you for the opportunity to review this manuscript and I am looking forward to receiving a modified version of the article.
Author Response
Thank you for reviewing our manuscript and your valuable and constructive comments.
Comment 1-Please explain why so many patients have been excluded from the study starting so early.
Response: Thank you for your insightful comments. We have added a justification for patient exclusions in the method.
We aimed to study older patients with BC and so, we included patients age ≥ 65 years with a diagnosis of BC and a BMI value (main variable of interest) in the dataset. To maintain a more homogeneous study population, we excluded patients with very short survival times (<6 months). This approach helps reduce potential confounding factors associated with patients who experience rapid mortality. For patients with multiple records, we selected the last survey as the baseline to ensure consistency in follow-up calculations for survival analysis.
Comment 2-Why has only the last survey for each patient been investigated?
Response: Thank you for your question.
Bladder cancer (BC) can recur after prior treatment, and for patients with recurrent BC diagnoses, earlier survey records—some dating back as far as 10 years—may not accurately reflect their most recent health status. Using the most recent survey helps capture the latest clinical information, making our analysis more relevant and reliable.
For survival analysis, it is essential to define a clear baseline time for calculating time-to-event outcomes. For patients with multiple records, we selected the last survey as the baseline to ensure consistency in follow-up calculations. We also obtained the KM survival when considering the first BC diagnose to assess if the results are consistent.
Comment 3- What was the average time the patients lived with bladder cancer before their inclusion in the study?
Response: Thank you for your question. On average, the time from the first bladder cancer (BC) diagnosis to the most recent survey (i.e., inclusion in the study) was 8.5 years (102 months) (Table 2).
However, it is important to note that BC can recur after initial treatment. For patients with recurrent BC, there may be periods during which they were disease-free between episodes of recurrence. As a result, we cannot precisely determine the continuous duration a patient lived with BC. Instead, our reported value reflects the time from initial diagnosis to study inclusion, irrespective of potential disease-free intervals.
Comment 4- As cancer represents a dynamic and often unpredictable disease, I believe the Authors should improve the materials and methods, as well as the results sections of the manuscript, particularly by offering greater transparency when it comes to the selection of the patients, as well as their baseline characteristics.
Response: Thank you for your thoughtful comment. We appreciate your feedback and have aimed to improve the clarity and completeness of the Materials and Methods and Results sections accordingly.
Studying rare diseases, such as cancer, presents challenges in assembling sufficiently large cohorts through prospective studies. Retrospective studies are often preferred in these cases because they allow researchers to leverage existing data, enabling larger sample sizes, cost-effective analyses, and quicker study completion—factors that are crucial for investigating infrequent conditions. For this study, we utilized the SEER-MHOS dataset, one of the most comprehensive resources available for analyzing cancer patients. To enhance transparency, we have added a detailed justification for patient exclusions in the manuscript and clarified the selection criteria and baseline characteristics. Although data has some limitations, our final cohort data are in line with prior studies both in terms of age, sex, and race.
The mean age of the sample set at the time of the survey is 77.68 years, with a standard deviation of 7.08 years. Given that the average age of cancer diagnosis is 73 years, and we only considering older population, a higher average age at the time of the survey was expected. Most participants are male (74.8%), aligning with data from the Moffitt Cancer Center, which indicates that bladder cancer is approximately four times more common in men than in women. The racial distribution of participants was also consistent with findings from other studies. The majority are White (85.6%), while 4.3% identified as Black or African American. Similarly, a study by Fang et al. (2020) reported a comparable racial distribution, with 89% White and 5.8% African American participants.
Comment 5- Please state the aim and objectives of your study more clearly at the end of the Introduction section.
Response: Thank you for your comment. The aims and objectives have been added at the end of Introduction section.
Thank you for again for your valuable suggestions.
Reviewer 3 Report
Comments and Suggestions for Authors
This study explores the association between BMI, survival, health-related quality of life and performance of ADLs in a cohort of older patients with bladder cancer based on SEER database. The comments are as follows:
1. In the methods part, the inclusion and exclusion criteria should be more specific, an estimation of sample size should also be explained.
2. In the duscussion part, the discussion of age, race, gender and other socioeconomic and health-related variables should be compared with other studies to see the consistency. That is, more literature citations should be added in discussion, even if some of them have been mentioned in the introduction part.
3. Besides study limitations, the authors need to summarize the strengths of the study.
4. All the tables should be as three-line format. The abbreviaitons in each tabale and figure should be given the full form in the notes. In Table 5, the references levels of variables shouls be added into each variables, not in notes, to make the table more clear.
5. The format of references should be unified, meeting with the standard/request of the journal.
Author Response
Thank you for reviewing our manuscript and your valuable and constructive comments.
Comment 1- In the methods part, the inclusion and exclusion criteria should be more specific, an estimation of sample size should also be explained.
Response: Thank you for your comment. We have added a detailed justification of the inclusion and exclusion criteria in the manuscript to enhance clarity.
Regarding sample size estimation, it is important to note that for large retrospective studies like ours, conducting a formal sample size estimation or post-hoc power analysis is generally not recommended. This approach is considered statistically problematic because power calculations based on already collected data remove the inherent randomness in sampling, potentially leading to misleading interpretations. The following reference support this rationale. Dziak JJ, Dierker LC, Abar B. The Interpretation of Statistical Power after the Data have been Gathered. Curr Psychol. 2020 Jun;39(3):870-877. doi: 10.1007/s12144-018-0018-1. Epub 2018 Oct 2. PMID: 32523323; PMCID: PMC7286546.)
Comment 2- In the discussion part, the discussion of age, race, gender and other socioeconomic and health-related variables should be compared with other studies to see the consistency. That is, more literature citations should be added in discussion, even if some of them have been mentioned in the introduction part.
Response: Thank you for your feedback. We have revised the discussion section. In the first paragraph we discuss the consistency of the sample cohort with what literature. In discussion section we have discussed the consistency of our finding regarding the demographic’s variables such as age, race and gender with other studies. More literature citation has been added.
The mean age of the sample set at the time of the survey was 77.68 years, with a standard deviation of 7.08 years. Given that the average age of cancer diagnosis is 73 years, and that the survey was conducted post-diagnosis for older patients, a higher average age at the time of the survey was expected. The majority of participants were male (74.8%), aligning with data from the Moffitt Cancer Center, which indicates that bladder cancer is approximately four times more common in men than in women. The racial distribution of participants was also consistent with findings from other studies. The majority were White (85.6%), while 4.3% identified as Black or African American. Similarly, a study by Fang et al. (2020) reported a comparable racial distribution, with 89% White and 5.8% African American participants.
The following references are added to the list of references:
Bladder Cancer statistics from MOFFIT Cancer Center.https://www.moffitt.org/cancers/bladder-cancer/faqs/bladder-cancer-statistics/
Fang, W., Yang, ZY., Chen, TY. et al. Ethnicity and survival in bladder cancer: a population-based study based on the SEER database. J Transl Med 18, 145 (2020). https://doi.org/10.1186/s12967-020-02308-w
Comment 3- Besides study limitations, the authors need to summarize the strengths of the study.
Response: Thanks for the comment. It has been added to the manuscript.
Comment 4-All the tables should be as three-line format. The abbreviations in each table and figure should be given the full form in the notes. In Table 5, the references levels of variables should be added into each variable, not in notes, to make the table clearer.
Response: Thanks for the comment. Tables format is revised. All figures and tables are revised with full form for labels with no abbreviation. Table 5 is revised and with all estimates including reference levels.
Comment 5-The format of references should be unified, meeting with the standard/request of the journal.
Response: Thanks for the comment. References are formatted according to MDPI reference guide.
Thank you again for valuable suggestions.
Round 2
Reviewer 1 Report
Comments and Suggestions for Authors
Thank you for the opportunity to re-review manuscript ID:
cancers-3524701.
The authors addressed all my comments correctly and in a quality manner. I believe that the introduced corrections contributed to making the work more clear and informative. Thanks to the authors.
Reviewer 3 Report
Comments and Suggestions for Authors
The authors have revised themanuscript according to the reviewer's comments. The format of the manuscript requires some revision in accordance with the requirements of the journal, e.g., the reference style. The authors need to check with it and revise it.